# Retinol-Binding Protein 4 and Visfatin Levels in Patients with Periodontitis and Obesity/Overweight: A Systematic Review and Meta-Analysis

Yuwei Zhang [1,2,†], Ru Jia [1,2,†], Yifei Zhang [1,2], Rui Zou [1,2], Lin Niu [1,2,3,*] and Shaojie Dong [1,2,3,*]

1 Key Laboratory of Shaanxi Province for Craniofacial Precision Medicine Research, College of Stomatology, Xi'an Jiaotong University, Xi'an 710004, China
2 Clinical Research Center of Shaanxi Province for Dental and Maxillofacial Diseases, Xi'an 710004, China
3 Department of Prosthodontics, College of Stomatology, Xi'an Jiaotong University, Xi'an 710004, China
* Correspondence: niulin@xjtu.edu.cn (L.N.); dongshaojie@xjtu.edu.cn (S.D.)
† These authors contributed equally to this work.

**Abstract:** Prior studies demonstrated an equivocal conclusion about the association between the level of retinol-binding protein 4 (RBP4)/visfatin and periodontitis patients with obesity. The aim of our study (Prospero ID: CRD42023469058) was to systematically review the available articles linking the biofluid levels of RBP4/visfatin to the comorbidity of periodontitis and obesity. Clinical trials were screened in accordance with specific inclusion criteria from seven databases up to November 2023. A quality assessment was performed with the Newcastle–Ottawa Scale and ROBINS-I tools for observational and interventional trials, respectively. The standard mean difference (SMD) with a 95% confidence interval (CI) related to the RBP4 level was recorded; the other indicators related to the visfatin level were measured via the mean difference (MD) with the corresponding 95% CI, and Fisher's Z transformation was measured to reveal the association using Review Manager 5.4. The current evidence was based on five observational studies and two interventional studies. All of them were included in the systematic review, and six of them were in the meta-analysis. Statistical analysis indicated that there was no significant difference in the circulating levels of RBP4 in the periodontitis patients with obesity or without, who were labeled as OP or NP, respectively (155 OP-107 NP: SMD = 1.38; 95% CI: −0.18–2.94, $p = 0.08$), as well as the periodontal healthy patients with a normal weight, who were labelled as NnP (116 OP-79 NnP: SMD = 6.76; 95% CI: −5.34–18.87, $p = 0.27$). Meanwhile, a significant higher level of serum visfatin was found in the OP patients than that of the NP (86 OP-45 NP: MD = 4.21; 95% CI: 2.65–5.77, $p < 0.00001$)/NnP (164 OP-88 NnP: MD = 13.02; 95% CI: 7.34–18.70, $p < 0.00001$) group. In addition, a positive association was observed between the serum RBP4 and body mass index/clinical attachment loss (CAL). And, then, there was a positive association between the serum visfatin and periodontal parameters, including the probing depth, CAL, and plaque index, as well as metabolic parameters, including the total cholesterol, triglycerides, fasting blood glucose, and low-density lipoprotein cholesterol. Here, the circulating RBP4 level was not independently related to the comorbidity of periodontitis and obesity, while serum visfatin was significantly associated with periodontitis and obesity. Notably, the positive association between circulating RBP4/visfatin and the periodontal parameters/metabolic parameters firmly suggested that the higher severity of the obese or periodontal status was associated with an elevated level of serum visfatin or RBP4 in the OP group. With more rigorous longitudinal research, the exact causations between RBP4/visfatin and the patients affected by obesity and periodontitis could be disentangled. RBP4 and visfatin might be novel, enlightening prospective bio-indexes for the targeted treatment of comorbidities.

**Keywords:** obesity; visfatin; RBP4; periodontitis; adipokine; meta-analysis

## 1. Introduction

Periodontitis is a multifactorial and chronic inflammatory disease, which is related to dysbiosis plaque biofilms and is characterized by the progressive destruction of the tooth-supporting apparatus [1–3]. The typical clinical features of periodontitis include the loss of periodontal tissue support, which is identified as bleeding during probing, gingival recession, the loss of clinical attachment, halitosis, bone tissue loss, and tooth mobility [4]. The current adopted classification scheme of *periodontal and peri-Implant diseases and conditions*, which is based on the characterization of the diseases using a multi-dimensional staging and grading system, followed the consensus report of work group two of the *2017 World Workshop* [3]. Furthermore, according to the current revised classification and other fundamental research, periodontitis is associated with the elevation in proinflammatory bioactive molecules, which could be a direct manifestation engaged in the development of systemic diseases, such as type 2 diabetes, cardiovascular disease, metabolic syndrome, and obesity [5–8].

Obesity, as another highly prevalent public health concern like periodontitis, is manifested as the accumulation of excess body adipose tissue [9]. Moreover, obesity is widely recognized as a status of systemic low-grade inflammation, and the interface between obesity and periodontitis could be connected to the inflammatory response, oxidative stress, and bone mass reduction at the periodontal and systematic levels [10–13]. While exploring the underlying mechanism, increased or decreased levels of certain inflammatory-related biomarkers are found to be deeply engaged in this process. Among them, adipokines, in addition to classic cytokines, such as tumor necrosis factor α (TNF-α), interleukin (IL-1β, IL-8, and IL-6), C-reactive protein (CRP), interferon-γ (IFN-γ), or chemoattractant protein-1 (MCP-1), are mostly secreted in adipose tissues from adipocytes, and adipokines are found to be released from periodontal tissues to regulate the inflammation process in the comorbidity of periodontitis and obesity [14,15]. So far, researchers have proved that leptin, resistin, visfatin, and retinol-binding protein 4 (RBP4) might exert proinflammatory effects, while adiponectin possesses anti-inflammatory characteristics. Intriguingly, the biofluid levels of these adipokines are altered in obesity/overweight and periodontitis states [16,17]. Biofluid plasma and serum samples are two classic, clinical, invasive ones; meanwhile, gingival crevicular fluid (GCF) from the deep or shallow periodontal pockets and saliva can be collected in a noninvasive way for analysis. In addition to identifying suspicious infectious diseases or hematologic disorders, we could envisage including adipokine analysis both at the baseline or follow-up timepoints for assessing the stage and grade of diagnosis and prognosis of periodontitis patients when it is a direct manifestation engaged in the development of obesity.

However, the prior studies demonstrated a rather equivocal conclusion about the association between the level of RBP4/visfatin and periodontitis patients with obesity [18,19]. To further investigate the role of RBP4 and visfatin as the path mechanistic link between periodontitis and obesity, we, thereby, carried out this systematic review and meta-analysis.

## 2. Methods

### 2.1. Protocol and Registration

This systematic review and meta-analysis proceeded in accordance with the preferred reporting items for systematic reviews and meta-analyses (PRISMA) guidelines [20], particularly the Meta-analysis of Observational Studies in Epidemiology guidelines [21], and was registered in the Prospective International Register of Systematic Reviews (PROSPERO, CRD42023469058), with the approval of all the authors.

### 2.2. Focused Question and Search Strategy

Our focused clinical question was constructed around the association between the RBP4/visfatin biofluid level and obese patients affected by periodontitis according to the PICO/PECO guidelines [22], primarily, "is there a difference in the RBP4/visfatin levels between periodontitis patients with obesity and those without obesity?", while the secondary

question was set as "is there a difference in the RBP4/visfatin levels between periodontitis patients with obesity and those without obesity and were periodontal healthy?"

### 2.2.1. Primary PECO Question

(Patient) Adult patients affected by periodontitis;
(Exposure) Obesity;
(Comparison) Normal weight as control;
(Outcome) Biofluid RBP4/visfatin levels.

### 2.2.2. Secondary PECO Question

(Patient) Adult patients;
(Exposure) Obesity and periodontitis;
(Comparison) Normal weight and periodontal healthy as control;
(Outcome) Biofluid RBP4/visfatin levels.

Subsequently, the corresponding systematic literature searching process was carried out from seven databases, including PubMed, Web of Science, the Cochrane Library, Scopus, ScienceDirect, ISI Web of Knowledge, and OpenGrey. The searches were limited to the first 200 hits in Google Scholar. Furthermore, a manual search of the relevant literature on the reference lists of the included original studies was performed. Two reviewers, YW. Z. and R. J., screened the controlled clinical trials that addressed the above-mentioned focused question in English from 2000 to November 2023. The search strategy included keywords related to ((obesity OR obese OR body mass index OR BMI) AND (periodontitis OR periodontal disease OR chronic periodontitis OR attachment loss OR adult periodontitis) AND (retinol-binding protein 4 OR visfatin OR nampt OR RBP4 OR adipokine)). Meanwhile, any kind of disagreement between the two reviewers was resolved by investigator SJ. D.

### 2.3. Inclusion and Exclusion Criteria

Two reviewers (YW. Z. and R. J.) performed the initial screening based on the titles and abstracts of the relevant articles. The full content of the preliminary included as well as the controversial articles were then carefully reviewed through a second screening. Discrepancies were resolved by a third reviewer (SJ. D.). To estimate the chance-adjusted inter-rater agreement, Cohen's κ- statistic was included and the κ-coefficient for this stage was 0.94, which indicated a substantial agreement.

Studies were considered eligible once they reached the following criteria. (1) The type of the study design was an observational study, including prospective studies, cross-sectional studies, and case-controlled or interventional studies with baseline levels presented. (2) The study reported the association between RBP4/visfatin levels and obese patients with peri-odontitis. (3) a, Obesity was the exposure and periodontitis patients with normal weight were the control group; b comorbidity of obesity and periodontitis was the exposure and periodontitis healthy and normal weight were the control group. Inclusion criteria for obesity and periodontitis are listed in Table 1. (4) The full text was published in English. (5) Participants did not have any known effects on periodontal health or any systemic diseases other than obesity, such as diabetes mellitus, orthodontic treatment, periodontal treatment within 6 months, women with pregnancy or lactating status, heavy smokers of more than 20 cigarettes per day, history of lipid-lowering therapy, chemotherapy or radio-therapy, intake of immunosuppressant, nonsteroidal anti-inflammatory drugs or antibiotics within 3 months, and less than 10 remaining teeth in the mouth. In addition, studies were also excluded if they were (1) in vitro or in vivo animal studies, review studies, case series or reports, conference abstracts, letters, and editor and expert comments; or (2) available only as an abstract without predefined outcome data necessary for further analysis (even though we tried to obtain original datasets from the corresponding authors). Inclusion and exclusion criteria mentioned above are listed in the Table S1 (Supplementary File).

**Table 1.** Characteristics of the included studies in the systematic review.

| Study | Design; Setting | Criterion for Inclusion | Sample and Methods | Case Group | | | Control Group | | |
|---|---|---|---|---|---|---|---|---|---|
| | | | | Sample Size | Average BMI (kg/m²) and Age (yrs) | Biomarkers | Sample Size | Average BMI (kg/m²) and Age (yrs) | Biomarkers |
| Martinez-Herrera et al., 2017 [23] | Interventional study; Spain | Obesity: Spanish Society for the Study of Obesity [24]; Obesity: body mass index (BMI) ≥ 30 kg/m²; Lean/normal: BMI ≤ 25 kg/m². Chronic periodontitis (CP): ≥4 teeth had ≥ 1 sites with probing depth (PD) ≥ 4 mm and clinical attachment loss (CAL) ≥ 3 mm [25] | Serum; nephelometry | Patients with obesity and periodontitis (OP) 96 male/female (m/f): 28/68 | OP Obesity BMI: (42.5 ± 2.1) Age: (42.7 ± 10.2) yrs. | Retinol-binding protein 4 (RBP4): (3.74 ± 1.12) mg/dL; tumor necrosis factor-alpha (TNF-α): (17.57 ± 9.91) pg/mL; interleukin (IL)-6: (3.72 ± 2.05) pg/mL | normal-weight patients and periodontal healthy (NnP) 64 m/f: 12/52; patients with obesity and periodontal healthy (OnP) 23 m/f: 3/20; normal-weight patients with periodontitis (NP) 48 m/f: 11/27 | NnP BMI: (21.8 ± 2.1) Age: (36.8 ± 11.7) yrs; OnP BMI: (39.6 ± 7.0) Age: (42.2 ± 11.1) yrs; NP BMI: (22.7 ± 1.9) Age: (39.4 ± 8.3) yrs; | NnP-RBP4: (3.06 ± 0.66) mg/dL; TNF-α: (7.72 ± 5.06) pg/mL; IL-6: (3.34 ± 2.78) pg/mL. OnP-RBP4: (3.21 ± 0.75) mg/dL; TNF-α: (12.41 ± 6.06) pg/mL; IL-6: (4.44 ± 2.71) pg/mL. NP-RBP4: (3.35 ± 0.70) mg/dL; TNF-α: (9.71 ± 4.36) pg/mL; IL-6: (4.06 ± 3.90) pg/mL. |
| Matern et al., 2020 [19] | Exploratory sub-analysis; prospective, randomized, double-blinded, parallel-group, multi-center ABPARO trial, Germany | Overweight: (Overweight: BMI ≥ 25 kg/m²; Lean/normal: BMI ≤ 25 kg/m². CP: ≥4 teeth had ≥ 1 sites with PD ≥ 4 mm and CAL ≥ 3 mm [25] | Plasma; ELISA | OP 40 m/f: 20/20 | OP Overweight BMI: 30.0 (28.9/32.3) Age: 54.0 (46.0/61.5) yrs. | RBP4: 31.5 (27.2/37.1) µg/mL; orosomucoid (ORM): 107.0 (89.0/123.0) mg/dL; high-sensitivity C-reactive protein (hsCRP): 1.5 (0.9/3.5) mg/L; Chemerin: 113.6 (86.3/125.0) ng/mL | NP 40 m/f: 20/20 | NP BMI: 22.9 (21.8/23.3) Age: 51.0 (42.5/59.5) yrs | RBP4: 32.7 (28.8/39.2) µg/mL; ORM: 79.7 (71.9/92.6) mg/dL; hsCRP: 1.4 (0.7/2.4) mg/L; Chemerin: 87.9 (73.0/103.3) ng/mL |
| Kanoriya et al., 2016 [26] | Single-center, cross-sectional, case-controlled study; India | Obesity: World Health Organization (WHO) guidelines [27] (Obesity: BMI ≥ 25 kg/m², Lean/normal: BMI 18.5–22.9 kg/m². CP: ≥4 teeth had ≥ 1 sites with PD ≥ 4 mm and CAL ≥ 3 mm, bone loss | GCF and serum; ELISA | OP 20 m/f: 9/11 | OP Obesity BMI: (29.09 ± 2.10) Age: (34.75 ± 6.01) yrs | Gingival crevicular fluid (GCF) RBP4: 23.50 ± 1.63; serum RBP4: 27.30 ± 1.80 ng/mL; GCF leptin: 132.00 ± 10.01; serum leptin: 375.55 ± 11.86 pg/mL | NnP 15 m/f: 8/7; OnP 15 m/f: 8/7; NP 20 m/f: 10/10 | NnP BMI: (20.44 ± 1.29) Age: (33.1 ± 6.28) yrs; OnP BMI: (28.18 ± 1.70) Age: (33.1 ± 6.28) yrs; NP BMI: (20.45 ± 1.32) Age: (35.10 ± 6.24) yrs; | NnP-GCF RBP4: (3.53 ± 1.06); serum RBP4: (6.53 ± 1.12) ng/mL; IL-6: (3.34 ± 2.78) pg/mL; GCF leptin: (231.33 ± 9.02); serum leptin: (81.07 ± 11.02) pg/mL; OnP-GCF RBP4: (9.00 ± 0.75); serum RBP4: (12.20 ± 1.37) ng/mL; IL-6: (3.34 ± 2.78) pg/mL; GCF leptin: (330.93 ± 8.31); serum leptin: (178.00 ± 12.64) pg/mL; NP-GCF RBP4: (16.75 ± 1.65); serum RBP4: (19.75 ± 1.51) ng/mL; GCF leptin: (33.00 ± 9.61); serum leptin: (272.70 ± 11.54) pg/mL |
| Jia et al., 2023 [28] | Cross-sectional study, case-controlled study; China | Obesity: WHO guidelines [29] (Obesity: BMI ≥ 30 kg/m²; Overweight: 30 ≥ BMI ≥ 25 kg/m², Lean/normal: 25 > BMI > 18.5 kg/m²). CP: the newest 2017 international classification of periodontitis [3] | GCF and serum; ELISA | OP 62 m/f: 32/30 | OP Obesity BMI: (35.80 ± 3.90) Age: (42.50 ± 7.42) yrs; Overweight BMI: (28.53 ± 3.21) Age: (39.49 ± 5.39) yrs | Serum visfatin: 8.87 ± 1.07; serum resistin: 76.35 ± 10.56 ng/mL | NnP 15 m/f: 5/10; OnP 14 m/f: 7/7; NP 21 m/f: 13/8 | NnP BMI: (22.89 ± 0.55) Age: (40.40 ± 8.32) yrs; OnP BMI: (32.30 ± 1.39) Age: (40.5 ± 7.33) yrs; NP BMI: (20.35 ± 0.58) Age: (39.60 ± 7.64) yrs. | NnP-serum visfatin: 3.26 ± 1.28; serum resistin: 13.23 ± 7.62; OnP-serum visfatin: 5.86 ± 1.29; serum resistin: 37.42 ± 10.59; NP-serum visfatin: 5.46 ± 1.32 ng/mL; serum resistin: 24.75 ± 8.37 |
| Li et al., 2018 [30] | Cross-sectional, case-controlled study; China | Obesity: Western Pacific Regional Office of WHO (WPRO) for obesity in adult Asians [31,32] (Obesity: BMI ≥ 25 kg/m²; Lean/normal: 25 > BMI > 18.5 kg/m²). CP: BOP, PD ≥ 3 mm, and CAL were presented in ≥ 1 site [4] | Serum ELISA | OP 78 Classified as [1] Mild (33), [2] Moderate (42) and [3] severe (13). | OP Obesity BMI ≥ 25 kg/m² | Visfatin: [1] 35.52 ± 4.19, [2] 40.84 ± 5.66, [3] 49.81 ± 5.57; leptin: [1] 12.89 ± 2.14, [2] 14.66 ± 2.37, [3] 17.39 ± 3.84; resistin: [1] 21.69 ± 3.43, [2] 29.64 ± 3.87, [3] 32.33 ± 4.06; adiponectin: [1] 7.64 ± 2.86, [2] 9.33 ± 3.24, [3] 9.89 ± 3.96 µg/L. | NnP 50; OnP 38 | NnP BMI (18.5, 25); OnP Obesity BMI ≥ 25 kg/m² | NnP-visfatin: 18.78 ± 2.72; leptin: 6.90 ± 1.58, resistin: 11.42 ± 2.41, adiponectin: 23.67 ± 3.94; OnP-visfatin: 33.27 ± 4.21; leptin: 12.37 ± 2.06, resistin: 21.48 ± 3.70, adiponectin: 8.83 ± 3.25 µg/L. |
| Doğan et al., 2022 [33] | Cross-sectional, case-controlled studies; Turkey | Obesity: WHO guidelines [29] (Obesity/overweight: BMI ≥ 25 kg/m², Lean/normal: BMI 18.5–25 kg/m²). CP: interdental CAL at ≥2 non-adjacent teeth or buccal or oral CAL ≥ 3 mm with PPD > 3 at ≥2 teeth [5] | Serum and saliva; ELISA | OP 24 | OP Obesity/overweight BMI: (29.87 ± 0.5) Age: (38.61 ± 1.24) yrs | Serum visfatin: 19 ± 0.95; Salivary visfatin: 23.5 ± 0.71 ng/mL | NnP 23; OnP 25; NP 24 | NP and NnP BMI: (22.09 ± 0.36) Age: (32.11 ± 1.11) yrs OnP Obesity/overweight BMI: (29.87 ± 0.5) Age: (38.61 ± 1.24) yrs | NnP-Serum visfatin: 11.5 ± 0.38; Salivary visfatin: 13.9 ± 0.67; OnP-Serum visfatin: 17.2 ± 0.88; Salivary visfatin: 19.8 ± 1.27; NP-Serum visfatin: 14 ± 0.99; Salivary visfatin: 19.7 ± 1.54 ng/mL. |

**Table 1.** *Cont.*

| Study | Design; Setting | Criterion for Inclusion | Sample and Methods | Case Group | | | Control Group | | |
|---|---|---|---|---|---|---|---|---|---|
| | | | | Sample Size | Average BMI (kg/m²) and Age (yrs) | Biomarkers | Sample Size | Average BMI (kg/m²) and Age (yrs) | Biomarkers |
| ÇETİNER et al., 2018 [34] | Interventional study; Turkey | Obesity: WHO guidelines [29] (Obesity: BMI ≥ 30 kg/m²; Lean/normal: BMI < 25kg/m²). CP: ≥30% of the sites with bone loss, and ≥2 nonadjacent teeth with ≥1 sites with PD ≥ 5 mm and CAL ≥ 5 mm in each quadrant, and positive bleeding on probing (BOP) | GCF; ELISA | OP 21 m/f: 0/21 | Obesity BMI: (34.76 ± 5.6) kg/m² OP Age: (44.67 ± 10.87) yrs | Visfatin: 21.53 ± 39.55; TNF-α: 9.00 ± 6.10; IL-6: 3.61 ± 4.43 Pg | NnP 10 m/f: 2/8; OnP 10 m/f: 1/9; NP 9 m/f: 7/2 | Lean/normal BMI: (23.53 ± 2.8) kg/m²; NnP-Age: (27.80 ± 3.12) yrs; OnP-Age: (46.50 ± 12.0) yrs; NP-Age: (44.22 ± 6.74) yrs | NnP-visfatin: 7.15 ± 3.12; TNF-α: 8.86 ± 0.87; IL-6: 0.00 ± 0; OnP -visfatin: 11.62 ± 10.71; TNF-α: 7.21 ± 9.77; IL-6: 1.32 ± 0.84; NP -visfatin: 10.65 ± 5.72; TNF-α: 10.54 ± 1.80; IL-6: 1.71 ± 2.01 pg. |

Patients with obesity/overweight and periodontitis (OP); normal-weight patients with periodontitis (NP); patients with obesity/overweight and periodontal healthy (OnP); normal-weight patients and periodontal healthy (NnP); chronic periodontitis (CP); probing depth (PD); clinical attachment loss (CAL); body mass index (BMI); World Health Organization (WHO); male/female (m/f); years (yrs); waist circumference (WC); gingival crevicular fluid (GCF); orosomucoid (ORM), high-sensitivity C-reactive protein (hsCRP), retinol-binding protein 4 (RBP4), interleukin (IL); bleeding on probing (BOP); tumor necrosis factor-alpha (TNF-α). Data were presented as median and inter-quartile range (25% quantile/75% quantile, IQR) or as mean (M) ± standard deviation (SD).

### 2.4. Data Extraction and Quality Assessment

Two reviewers (YW. Z. and R. J.) independently performed data extraction. Information from the included studies was tabulated according to the first author, publication year, study design, setting, participant characteristics, inclusion criteria for periodontitis and obesity, sample and method, and sample characteristics with investigated cytokines/adipocytokine baseline levels. Data were collected based on the question of interest in the systematic review. The reviewers cross-checked all extracted data, and any dissent was resolved by discussion until consensus was reached. The Kappa score was 0.95. The selected studies were assessed for quality using the Newcastle–Ottawa Scale for included observational studies [19,26,28,30,33] and Cochrane-advocated ROB-2 tool for evaluating the risk of bias in RCTs [35,36]. For assessing risk of bias of non-RCTs, the ROBINS-I tool was used by looking into preintervention, intervention, and postintervention domains. Studies were evaluated as low, moderate, serious, and critical risk of bias, from low (less than 1 moderate concern in the included domains; comparable) to critical risk of bias (critical concerns/multiple serious concerns; unreliable). The specific judgment criteria are shown in Table S2 [37].

### 2.5. Statistical Analysis

Biomarker data were expressed as mean (M) $\pm$ standard deviation (SD). Mean difference (MD) and the corresponding 95% confidence interval (95% CI) were used after the order of magnitude was approaching unit unification. Otherwise, standard mean difference (SMD) and the corresponding 95% CI were calculated to estimate the differences in RBP4/visfatin levels between groups. Furthermore, we evaluated the association between circulating serum levels of RBP4/visfatin and OP, serum RBP4 level and body mass index (BMI) probing depth (PD), and clinical attachment loss (CAL). We also evaluated the association between serum visfatin level and PD, CAL, visible periodontal plaque index (PI), total cholesterol (TC), triglycerides (TG), fasting blood glucose (FBG), glycosylated hemoglobin (HbA1c), low-density lipoprotein cholesterol (LDL-C), and high-density lipoprotein cholesterol (HDL-C), respectively. The Pearson correlation coefficients of related studies were recorded and Fisher's Z transformation was utilized to estimate a mean transformed correlation weighted by each sample size given by the data in our included studies.

The significance level of the overall effect was determined by Z test. Forest plots were illustrated for the effect size and corresponding 95% CI. Based on Cochran Q statistics, $I^2$ statistic was used to estimate between-study heterogeneity [38,39]. If $I^2 > 50\%$, substantial heterogeneity was assumed and a random effect model was used. Otherwise, a fixed effect model was applied. Due to the limitation of included studies in the subgroup analysis, publication bias could not be assessed using funnel plot. All statistical analyses were performed using Review Manager (RevMan) 5.4 statistical software, which was provided by Cochrane Collaboration. The significance *p* value was set at 0.05.

## 3. Results

### 3.1. Literature Search

A total of 152 studies were identified from the search strategy in seven electronic databases and one manuscript was found by manual searching. After screening the titles and abstracts, 129 studies that were considered duplicates and 3 studies that were considered irrelevant studies were excluded by two reviewers independently. Then, the full text of the remaining 21 articles was assessed. A total of 14 studies were excluded because they did not meet the inclusion criteria and the reasons are listed in Figure S1. Finally, seven studies were considered eligible for inclusion in this systematic review and six of them were included in the meta-analysis. The flowchart presenting the above process is displayed in Figure 1.

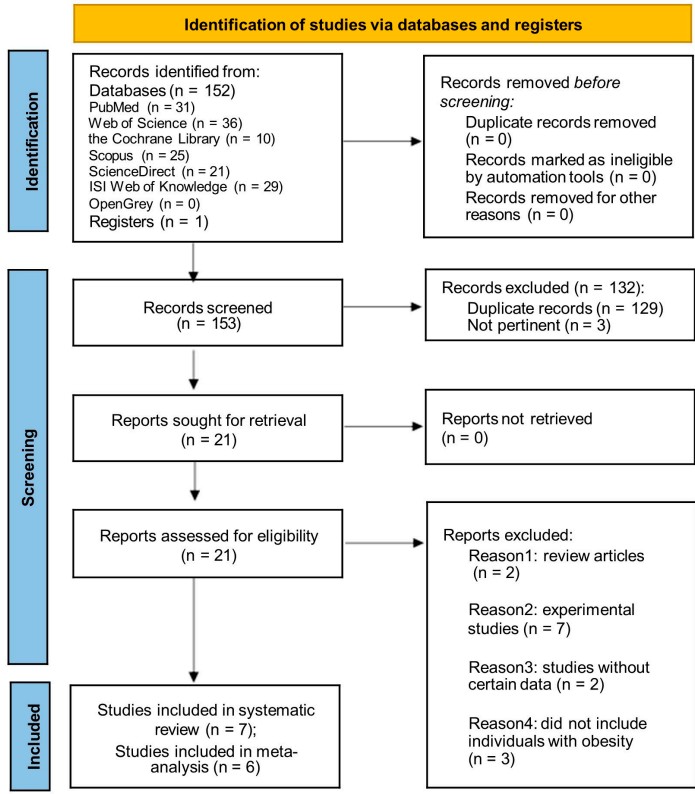

**Figure 1.** PRISMA 2020 flow diagram presenting the literature search and the selection phases of the incorporated articles of the systematic review and meta-analysis [40].

### 3.2. Characteristics and Quality Assessment of Study

The characteristics of the included seven studies published between 2016 and 2023 are presented in Table 1. The sample size of these studies ranged from 50 to 231. Among the seven incorporated studies, three of them were set in Asia (China [28,30]; India [26]); meanwhile, two of them were set in Europe (Spain [23]; Germany [19]) and two of them were set in Turkey, Asia, and Europe [33,34], which reflected the relatively diverse ethnicities. Five of the included seven studies were observational studies [19,26,28,30,33]. Most of them were designed as a typical case–control form, while one study [34] was set as an interventional study as the effect of nonsurgical periodontal treatment. Calorie-restricted diet therapy was assessed through baseline and follow-up biomarker levels. The data of baseline level were recorded. All manuscripts were published in English.

As for the definition of obesity criteria in included participants, all studies followed BMI thresholds with or without anthropometric or metabolic measures, such as waist circumference (WC), waist-to-hip ratio, or body fat, which were considered to identify obesity, overweight, and normal weight. And most studies set these cut-off points (Obesity: BMI $\geq$ 30 kg/m$^2$; Overweight: 30 $\geq$ BMI $\geq$ 25 kg/m$^2$; Lean/normal: 25 > BMI kg/m$^2$) [29]. Meanwhile the population in adult Asians [26,30] followed the Western Pacific Regional Office of WHO (WPRO) for obesity criteria [27,31,32] (Obesity: BMI $\geq$ 25 kg/m$^2$; Lean/normal: 25 > BMI > 18.5 kg/m$^2$). In addition, various definitions of periodontitis were used by different authors, mostly following the criteria ($\geq$4 teeth had $\geq$ 1 site with PD $\geq$ 4 mm and CAL $\geq$ 3 mm, or with bone loss) [25]. All of them were accepted for the purposes of this study due to the differences in region and year of publication, while the latest 2017 World Workshop [3] was preferred. Biofluid samples, such as circulating serum and plasma, GCF, and saliva, were collected in different studies for analysis through enzyme-linked immunosorbent assay (ELISA), except for one study using the methodology as nephelometry [23]. As for the statistical analysis in meta-analysis, only circulating serum and plasma levels were taken into account due to the number of articles.

The quality assessment of the included five observational studies using the Newcastle–Ottawa Scale and Cochrane-advocated ROBINS-I tools for two interventional articles are shown in Tables S2 and S3. The low risk of bias of non-RCTs and high scores of observational studies ranging from 7 to 8 stars indicated credible quality (Tables S2 and S3).

### 3.3. Subgroup Meta-Analysis

Sixteen subgroup analyses were conducted according to the category of biomarker (RBP4/visfatin) and types of PIEO investigations. Subgroup meta-analyses were performed using SMD and associated 95% CI in the analysis of RBP4 in patients with OP. MD and 95% CI were used in the analysis of baseline serum levels of visfatin in patients with OP. Fisher's Z transformation was utilized to estimate a mean transformed correlation weighted by each sample size given by the data in our included studies.

Primary investigations were around "is there a difference in the RBP4/visfatin levels between periodontitis patients with obesity and those without obesity?" To answer this question, subgroup analysis was also first performed based on the population between patients with OP and normal-weight patients with periodontitis (NP) (Figures 2 and 3). The random effect model was selected because substantial heterogeneity was observed ($I^2 > 50\%$). No significant difference in circulating RBP4 level was observed among participants with and without periodontitis [19,23,26] (Figure 2, SMD = 1.38; 95% CI: −0.18–2.94, $p = 0.08$). In addition, a significant difference in serum visfatin level was found between two groups of periodontitis participants with and without obesity, with a higher level of serum visfatin found in OP group (Figure 3, MD = 4.21; 95% CI: 2.65–5.77, $p < 0.00001$) [28,33].

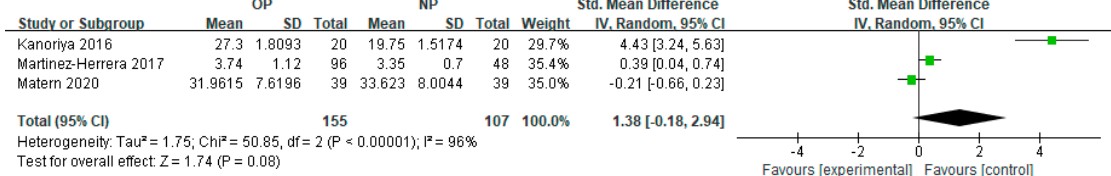

**Figure 2.** Subgroup analysis comparing baseline circulating levels of RBP4 within patients with obesity/overweight and periodontitis (OP) and normal-weight patients with periodontitis (NP) [18,19,26].

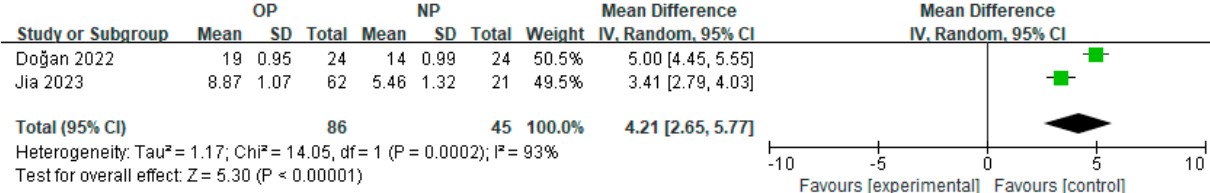

**Figure 3.** Subgroup analysis comparing baseline serum levels of visfatin within patients with obesity and periodontitis (OP) and normal-weight patients with periodontitis (NP) [28,30,33].

Meanwhile, the secondary question was set as "is there a difference in the RBP4/visfatin levels between periodontitis patients with obesity and those without obesity and were periodontal healthy?" The random effect model was selected and, in the included two studies, no significant difference in serum RBP4 level [23,26] was found between the groups of obese participants with periodontitis and systematic healthy participants without periodontal diseases (NnP) (Figure 4, SMD = 6.76; 95% CI: −5.34–18.87, $p = 0.27$). Furthermore, a significant difference was found in serum visfatin level between the groups of obese participants with periodontitis and systematic healthy participants without periodontal diseases A higher level of serum visfatin was found in the OP group than NnP (Figure 5, MD = 13.02; 95% CI: 7.34–18.70, $p < 0.00001$) [28,30,33].

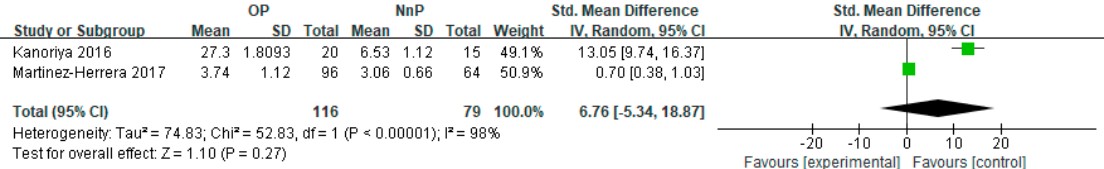

**Figure 4.** Subgroup analysis comparing baseline serum levels of RBP4 within patients with obesity and periodontitis (OP) and normal weight and periodontal healthy participants (NnP) [18,26].

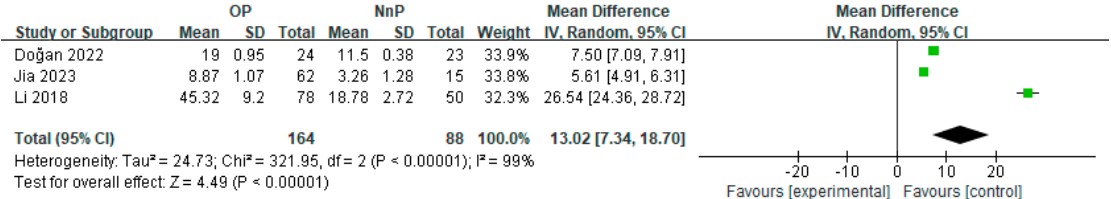

**Figure 5.** Subgroup analysis comparing baseline serum levels of visfatin within patients with obesity and periodontitis (OP) and normal weight and periodontal healthy participants (NnP) [28,30,33].

Furthermore, to explore the association between RBP4/visfatin and periodontal and obesity parameters, Fisher's Z transformation was selected to calculate the given Pearson correlation coefficient. BMI is the most important indicator of obesity status. Regarding the association between serum RBP4 and BMI values in group OP, a fixed effect model was chosen ($I^2 = 47\%$, $p = 0.17$). The result (Figure S1) revealed a significant positive correlation of serum RBP4 and BMI, with a Fisher's Z of 0.70 (95% CI: 0.50 to 0.91, $p < 0.00001$). Moreover, Figures S2 and S3 showed a rather positive correlation between serum RBP4 and probing depth (PD) or clinical attachment loss (CAL) (PD: Fisher's Z of 0.02, 95% CI: −0.01 to 0.41, $p = 0.06$; CAL: Fisher's Z of 0.22, 95% CI: 0.01 to 0.43, $p = 0.04$). As for the association between serum visfatin and periodontal parameters in the OP group, forest plots indicated a positive correlation between serum visfatin and three periodontal parameters (Figures S4–S6, PD: Fisher's Z of 0.59, 95% CI: 0.16 to 1.03, $p = 0.08$; CAL: Fisher's Z of 0.59, 95% CI: 0.21 to 0.98, $p = 0.02$; PI: Fisher's Z of 0.41, 95% CI: 0.19 to 0.63, $p = 0.05$). In order to explore the association between serum visfatin and obese parameters, further investigation was proceeded. Forest plots indicated a positive correlation of serum visfatin and four obesity parameters (Figures S7–S12, TC: Fisher's Z of 0.93, 95% CI: 0.00 to 1.86, $p = 0.05$; TG: Fisher's Z of 1.20, 95% CI: 0.76 to 1.64, $p < 0.00001$; FBG: Fisher's Z of 0.58, 95% CI: 0.22 to 0.94, $p = 0.002$; LDL-C: Fisher's Z of 0.89, 95% CI: 0.72 to 1.06, $p < 0.00001$) and no apparent association in two others (HbA1c: Fisher's Z of 0.15, 95% CI: −0.02 to 0.32, $p = 0.09$; HDL-C: Fisher's Z of −0.50, 95% CI: −1.53 to 0.53, $p = 0.34$).

## 4. Discussion

To date, the relationship between RBP4 and periodontitis in patients with or without obesity still remains controversial. Furthermore, we have not yet been able to conclude whether the biofluid RBP4 levels, particularly the circulating levels, of obese patients suffering from periodontitis significantly differ from those of systematically healthy patients with periodontitis or even normal-weight patients without periodontitis. Previous studies have shown either decreased or increased circulating levels of RBP4 in obese patients with periodontitis compared to those with normal weight [19,23,26]. However, at a methodological level, due to the discrepancy in age, ethnicity, gender, pre-analytical conditions (i.e., fasting time), samples, inconsistent measurement methods (essentially different types of ELISA kits/nephelometry), physical conditions, etc., the actual detected circulating levels of RBP4 varied from one to another. The combination of these confounding factors could result in heterogeneity, which explains why SMD in a discrepancy situation was applied. Therefore, based on the provided data [23,26], we further focused on specific groups of patients with obesity and periodontitis to analyze the correlation between serum

RBP4 and several metabolic or periodontal parameters through Fisher's Z. Intriguingly, we found a positive association between serum RBP4 and BMI/CAL, which indicated that an increased level of serum RBP4 is associated with a higher degree of BMI value or CAL figure, as it was widely known that BMI or CAL was the most basic index reflecting the degree of obesity or the severity of periodontitis [29,41].

Regarding the potential role of RBP4 in the comorbidity of periodontitis and obesity, previous research from our study [16] or others has suggested that increased RBP4 (systematically or periodontal expression) level might be a novel contribution to the inflammatory process of periodontitis among obese participants. Effective nonsurgical periodontal therapy has been shown to reduce serum RBP4 levels at a 3-month follow-up timepoint in patients with both obesity and periodontitis. Considering the possible cause, it is known that RBP4 was initially identified as a member of the lipocalin family, which binds to vitamin A and transthyretin that could be secreted into the circulation and widely known as a predictor of atherosclerosis, endothelial dysfunction, and other cardiovascular diseases. Furthermore, RBP4 has strong associations with obesity and periodontitis due to its specific relationship with inflammation and oxidative stress. The comorbidity status could break the dynamic equilibrium between reactive oxygen species and antioxidant scavenging defense system, which lead to increased oxidative stress and exacerbated aggravated inflammatory response, particularly in the oral environment [10,18,42].

Additionally, we identified significantly elevated serum visfatin level in obese patients suffering from periodontitis compared to the NP and NnP groups. This suggests that visfatin could serve as a potential biomarker for diagnostics, therapeutic judgmental reference, and clinical treating target. Furthermore, there was a positive correlation between serum visfatin levels and periodontal parameters (PD, CAL, and PI) as well as metabolic parameters (TC, TG, FBG, and LDL-C), which firmly suggested that the higher severity of obese or periodontal status were associated with a higher level of serum visfatin in the OP group. It is proposed that visfatin, serving as a proinflammatory index and immunomodulator for periodontitis, could subsequently stimulate the synthesis of systematic inflammatory mediators and proteases while inhibiting the apoptosis of inflammatory cells [14,43,44]. Our meta-analysis results corroborate that visfatin expression increases in human patients' gingival tissues affected by both aggressive and chronic periodontitis [44].

In summary, our systematic review and meta-analysis presented more convincing details and comprehensive analysis about the association between RBP4/visfatin biofluid level and patients with periodontitis who are obese. However, due to the limited number of relevant articles, which were mostly observational/cross-sectional studies, further improvement is indispensable for exploring the causal relationship. Basically, the overall heterogeneity above was somehow unavoidable due to the clinical heterogeneity, including physical conditions of subjects, sample size, and clinical methodology (i.e., discrepancy in age, ethnicity, gender, preanalytical conditions, samples category, sample collection, and inconsistent measurement methods). All these factors could lead to a conflicting result and reduce reliability. In addition, since articles without English publication or incomplete data had been excluded, publication bias was inevitable. If the sample size was adequate, stratification treatment based on confounding factors mentioned above would be a better-quality control option. Meanwhile, according to the quality assessment results of two interventional [23,34] studies based on the ROBINS-I tool, most domains were considered as low risk and moderate risk when confronting the selection of participants (unmentioned continuous recruitment of patients) or missing data (dropouts of one study [23] were more than 5%). Meanwhile, regarding five observational studies [19,26,28,30,33] using the Newcastle–Ottawa Scale tool, the selection and exposure of participants (unmentioned continuous recruitment of patients and discrepancy in BMI thresholds for obesity definition due to ethnic groups) were main concerns that affected the quality. Basically, we only extracted the baseline levels of targeted biomarkers. However, our findings should be interpreted with caution, even if, so far, the overall quality assessment was beyond moderate because of underlying risks and clinical heterogeneity.

In summary, it is meaningful and worthwhile to expect a deeper and therapeutic investigation of RBP4/visfatin in obese patients with periodontitis. According to the currently available evidence from clinical studies, laboratory animal science, and in vitro studies, visfatin and RBP4 may exert proinflammatory effects and there is a significant positive correlation between the level of visfatin/RBP4 and the degree of alveolar bone resorption in patients with obesity and periodontitis. Further research should focus on revealing the role and mechanism of RBP4/visfatin in regulating bone defects in obese periodontitis subjects in vivo. Then, the evaluation of the feasibility of treating alveolar bone resorption in obese patients with periodontitis with RBP4/visfatin monoclonal antibodies or blockers (such as combining novel drug delivery platforms, like injectable gels or microneedles) lays the foundation for long-term clinical transformation and precise treatment.

## 5. Conclusions

To sum up, our systematic review incorporated seven high-quality observational/interventional studies. The meta-analysis revealed that the circulating levels of RBP4 in patients with periodontitis and obesity were not significantly different from those in periodontally healthy patients. There was also a positive association between serum RBP4 and BMI/CAL. Moreover, patients with periodontitis and obesity had significantly higher levels of serum visfatin than those without obesity and periodontally healthy patients. Additionally, there was a positive association between serum visfatin and periodontal parameters (PD, CAL, and PI) as well as metabolic parameters (TC, TG, FBG, and LDL-C). These findings strongly suggest that the severity of obesity or periodontal status is associated with increased levels of serum visfatin or RBP4 in the OP group. With further rigorous longitudinal research, the exact causal relationships between RBP4/visfatin and patients affected by obesity and periodontitis can be determined. RBP4 and visfatin may serve as novel prospective biomarkers of the comorbidity from both diagnostic and prognostic perspectives. Furthermore, assessing the feasibility of treating alveolar bone resorption in obese patients with periodontitis using RBP4/visfatin monoclonal antibodies or blockers lays the groundwork for long-term clinical transformation and precise treatment.

**Supplementary Materials:** The following supporting information can be downloaded at: https://www.mdpi.com/article/10.3390/cimb45120614/s1, Figure S1: Subgroup analysis of the association of serum RBP4 levels with body mass index (BMI) in patients with obesity and periodontitis (OP); Figure S2: Subgroup analysis of the association of serum RBP4 levels with probing depth (PD) in patients with obesity and periodontitis (OP); Figure S3: Subgroup analysis of the association of serum RBP4 levels with clinical attachment loss (CAL) in patients with obesity and periodontitis (OP); Figure S4: Subgroup analysis of the association of serum visfatin levels with probing depth (PD) in patients with obesity and periodontitis (OP); Figure S5: Subgroup analysis of the association of serum visfatin levels with clinical attachment loss (CAL) in patients with obesity and periodontitis (OP); Figure S6: Subgroup analysis of the association of serum visfatin levels with visible periodontal plaque index (PI) in patients with obesity and periodontitis (OP); Figure S7: Subgroup analysis of the association of serum visfatin levels with total cholesterol (TC) levels in patients with obesity and periodontitis (OP); Figure S8: Subgroup analysis of the association of serum visfatin levels with triglycerides (TG) levels in patients with obesity and periodontitis (OP); Figure S9: Subgroup analysis of the association of serum visfatin levels with fasting blood glucose (FBG) levels in patients with obesity and periodontitis (OP); Figure S10: Subgroup analysis of the association of serum visfatin levels with glycosylated hemoglobin (HbA1c) levels in patients with obesity and periodontitis (OP); Figure S11: Subgroup analysis of the association of serum visfatin levels with low density lipoprotein cholesterol (LDL-C) levels in patients with obesity and periodontitis (OP); Figure S12: Subgroup analysis of the association of serum visfatin levels with high density lipoprotein cholesterol (HDL-C) levels in patients with obesity and periodontitis (OP). Table S1: Inclusion and exclusion criteria and studies excluded after full-text analysis and related reasons; Table S2: Risk of bias assessment of included studies according to the ROBINS-I tool; Table S3: Quality assessment of incorporated studies using the Newcastle–Ottawa Scale tool.

**Author Contributions:** S.D., R.J. and Y.Z. (Yuwei Zhang) conceived and visualized the article. Y.Z. (Yuwei Zhang), Y.Z. (Yifei Zhang) and R.J. carried out the review. Y.Z. (Yuwei Zhang), R.J. and Y.Z. (Yifei Zhang) analyzed the data and wrote the original draft. S.D., R.Z. and L.N. reviewed and edited. All authors have read and agreed to the published version of the manuscript.

**Funding:** This work is supported by Fundamental Research Funds of Xi'an Jiaotong University for Free Exploration and Innovation—Project for Teacher (XZY012021069) and Project to Enhance the Base of Innovation Ability of Xi'an City—Medical Research (21YXYJ0123).

**Institutional Review Board Statement:** Not applicable.

**Informed Consent Statement:** Not applicable.

**Data Availability Statement:** The datasets used and/or analyzed during the study are available from the corresponding author upon reasonable request.

**Conflicts of Interest:** The authors declare no conflict of interest.

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
