# Peer review of "Retinol-Binding Protein 4 and Visfatin Levels in Patients with Periodontitis and Obesity/Overweight: A Systematic Review and Meta-Analysis"

_cimb, doi:10.3390/cimb45120614_

Round 1
Reviewer 1 Report
Comments and Suggestions for Authors
This paper aims to systematically review the available articles linking biofluid levels of RBP4/visfatin to the comorbidity of periodontitis and obesity.
It is a well-structured and organized document. After a general look of the paper I would suggest :
Abstract should include some results from the flowchart and an indication of number os specimens. Abstract should not include non described acronyms
Introduction – it can be more developed to integrate this determination in day to day clinical appointments.
Line 73- repetition of acronym´s description. Please insert the explanation and acronym and then use it in all the rest of the text.
Table 1 should be horizontally orientated.
Discussion- limitations of the study should be pointed out. Also, an overview of how the information of the text could help in the clinical treatment of periodontal diseases (e-g- gels, etc)
Author Response
Reviewer:
This paper aims to systematically review the available articles linking biofluid levels of RBP4/visfatin to the comorbidity of periodontitis and obesity.
It is a well-structured and organized document. After a general look of the paper, I would suggest:
- Abstract should include some results from the flowchart and an indication of number of specimens. Abstract should not include non-described acronyms.
Response: Thank you so much for the kind encouragement and valuable comment, which really mean a lot to us. We have added the information in the manuscript as “Methods: Clinical trials were screened in accordance with the specific inclusion criteria from seven databases up to December 2022. Quality assessment was performed with Newcastle-Ottawa Scale tool and ROBINS-I tools for observational and interventional trials, respectively. Standard mean difference (SMD) with 95% confidence interval (CI) related to RBP4 level were recorded, other indicators related to visfatin level were measured via mean difference (MD) with corresponding 95% CI, and Fisher’s Z transformation was measured to reveal the association via Review Manager 5.4.
Results: The current evidence was based on the 5 observational studies and 2 interventional studies. All of them were included in the systematic review and six of them were in the meta-analysis. Statistical analysis indicated that there was no significant difference in circulating levels of RBP4 between periodontitis patients with (OP) and without (NP) obesity (155 OP-107 NP: SMD ꞊ 1.38; 95% CI: -0.18 – 2.94, p ꞊ 0.08)/normal weight and periodontal healthy patients, which labelled as NnP(116 OP-79 NnP: SMD ꞊ 6.76; 95% CI: -5.34–18.87, p ꞊ 0.27). Meanwhile, a significant higher level of serum visfatin was found in OP patients than that of NP (86 OP-45 NP: MD ꞊ 4.21; 95% CI: 2.65 – 5.77, p < 0.00001)/NnP (164 OP-88 NP: MD ꞊ 13.02; 95% CI: 7.34 – 18.70, p < 0.00001) group. Besides, a positive association was observed between the serum RBP4 and body mass index /clinical attachment loss (CAL). And then, there was a positive association between serum visfatin and periodontal parameters including probing depth, CAL, plaque index as well as metabolic parameters including total cholesterol, triglycerides, fasting blood glucose, and low density lipoprotein cholesterol.” (page 2-3, line 20-40)
- Introduction – it can be more developed to integrate this determination in day-to-day clinical appointments.
Response: Thank you very much for your constructive reminding. We have added the information in the manuscript according to your advice. Please read the details in the revised version as “Biofluid samples such as plasma and serum are two classic clinical invasive ones, meanwhile, gingival crevicular fluid (GCF) from the deep or shallow periodontal pockets and saliva could be collected in a non-invasive way for analysis. Besides identifying suspicious infectious diseases or hematologic disorders, we could envisage including adipokines analysis both at the baseline or follow-up timepoints for assessing the stage and grade of diagnosis and prognosis in periodontitis when it is a direct manifestation engaged in the development of obesity.” (page 3, line 79-85)
- Line 73- repetition of acronym´s description. Please insert the explanation and acronym and then use it in all the rest of the text.
Response: We are deeply sorry for the repetition of acronym (RBP4) and deleted the second one in the revised version (page 3, line 86-87).
- Table 1 should be horizontally orientated.
Response: Thank you so much for your sincere suggestion, it is so helpful and we have changed the table into horizontally orientated in the updated version.
- Discussion- limitations of the study should be pointed out. Also, an overview of how the information of the text could help in the clinical treatment of periodontal diseases (e-g- gels, etc.)
Response: We appreciate the reviewer for the constructive suggestion. We have added the information in the manuscript as
“However, given by the limited related articles, which were mostly observational/cross-sectional studies, further improvements were indispensable for exploration on the cause-effect relationship. Basically, the overall heterogeneity stated above was somehow unavoidable due to the clinical heterogeneity, including physical conditions of subjects, sample quantity, and clinical methodology (i.e., the aforementioned discrepancy in age, ethnicity, gender, pre-analytical conditions, samples category, sample-collecting and inconsistent measurement methods). All these factors could lead to a conflicting result and reduce the reliability. What’s more, since articles without English publication or incomplete data had been excluded, publication bias was inevitable. If the sample size was sufficient, stratified treatment based on the confounding factors mentioned above would be a better-quality control option. And according to the quality assessment results of two interventional [33, 34] studies based on the ROBINS-I tool, most of domains were considered as low risk and moderate risk when confronting the selection of participants (unmentioned continuous recruitment of patients) or missing data (dropouts of one study [33] were more than 5%). Meanwhile, as for five observational studies [19, 23-26] using the Newcastle-Ottawa Scale tool, the selection and exposure of participants (unmentioned continuous recruitment of patients and discrepancy in BMI threshold for obesity definition due the ethnic groups) were the main concerns that affected the quality. Basically, we only extracted the baseline level of the aimed biomarkers, however, our findings should be interpreted with caution even if so far the overall quality assessment was beyond moderate since the underlying risks and clinical heterogeneity.
All in all, expectance on a deeper and therapeutic investigation of RBP4/visfatin in obese patients affected by periodontitis is meaningful and worthwhile. Given by the currently available evidence from clinical studies, laboratory animal science and in vitro studies, visfatin and RBP4 might exert pro-inflammatory effects, and there is a significant positive correlation between the level of visfatin/RBP4 and the degree of alveolar bone absorption in patients with obesity & periodontitis. Further studies should focus on revealing the role and mechanism of RBP4/visfatin in regulating bone defects in obese periodontitis subjects in vivo. Then, the evaluation of the feasibility of treating alveolar bone absorption in obese patients with periodontitis with RBP4/visfatin monoclonal antibodies or blockers (such as combining novel drug delivery platforms, like injectable gels or microneedles) lays the foundation for long-term clinical transformation and precise treatment.” (page 17-18, line 357-387).
In the end, we would like to express our special thanks to you for all your comments and suggestions, which are of great help to improve the quality of our manuscript!
Reviewer 2 Report
Comments and Suggestions for Authors
I reviewed the manuscript “Retinol-binding protein 4 and visfatin level in patients with periodontitis and obesity/overweight: a systematic review and meta-analysis”, the purpose of which was to review available articles linking biofluidic levels of RBP4/visfatin to the comorbidity of periodontitis and obesity.
I congratulate the Authors on the clarity and methodology of the manuscript. Below are some suggestions for each section:
Abstract:
- Line 31: there is one parenthesis too many.
M&M:
- Six databases were indicated in the abstract and seven in the M&Ms.
- Were the patients considered for the first question also adults? Or were cases of periodontitis in young people also considered if found?
Result:
- Use the PRISMA 2020 flow diagram and indicate the number of items found for each database.
- Line 206: report the references of the five studies.
- I would suggest arranging the layout with the tables horizontally to make it more streamlined and easier to read.
- Add all abbreviations in the table, even if they were already written in the text (e.g. OP, CP, NnP, OnP, NP, RBP, IL, BMI and check the rest).
Discussion:
- Add to the discussion a comment on the quality of the included studies, most of which were judged to be of low quality.
Conclusion:
- What are the future prospects for clinicians based on these results?
References:
- References in the text are positioned as an index, thus not formatted according to the journal guidelines. Check also the format of the references on line 312.
- References in the bibliography are not formatted according to journal guidelines (e.g. year in bold, volume and journal in italics). Check them all.
Author Response
Reviewer:
I reviewed the manuscript “Retinol-binding protein 4 and visfatin level in patients with periodontitis and obesity/overweight: a systematic review and meta-analysis”, the purpose of which was to review available articles linking biofluidic levels of RBP4/visfatin to the comorbidity of periodontitis and obesity.
I congratulate the Authors on the clarity and methodology of the manuscript. Below are some suggestions for each section:
- Abstract:
Line 31: there is one parenthesis too many.
Response: Thank you so much for your valuable suggestion. We have made certain modification of the sentence to be “Statistical analysis indicated that there was no significant difference in circulating levels of RBP4 of periodontitis patients with obesity or not, labeled as OP or NP respectively, (155 OP-107 NP: SMD ꞊ 1.38; 95% CI: -0.18 - 2.94, p ꞊ 0.08), as well as periodontal healthy patients with normal weight, which labelled as NnP (116 OP-79 NnP: SMD ꞊ 6.76; 95% CI: -5.34 - 18.87, p ꞊ 0.27).” (Page 2, line 29 - 33)
- M&M:
2.1. Six databases were indicated in the abstract and seven in the M&Ms.
Response: We are deeply sorry for the discrepancy and changed as unified seven databases “Clinical trials were screened in accordance with the specific inclusion criteria from seven databases up to November 2023.” (page 1, line 20-21)
2.2. Were the patients considered for the first question also adults? Or were cases of periodontitis in young people also considered if found?
Response: We are deeply apologized for the misunderstanding in the text, and we have modified it into the all the participants were adult, for the WHO standard for obesity or overweight we used to be set for adults (For adults, normal-weight, overweight, and obesity are defined as BMI 18.5 to <25 kg/m2, BMI 25 to <30 kg/m2, and BMI ≥ 30 kg/m2, respectively). Besides, juvenile periodontitis as a special type of periodontitis, could be developing rapidly, unlike traditional chronic periodontitis.
- Result:
-3.1. Use the PRISMA 2020 flow diagram and indicate the number of items found for each database.
Response: Thank you so much for your sincere suggestion. We have changed it into the PRISMA 2020 version in Page 8.
3.2. Line 206: report the references of the five studies.
Response: We appreciate the reviewer for the constructive suggestion. We have reported the five studies in this revision as “Five of the included seven studies were observational studies [19, 23-26]” (page 8, line 214-215).
3.3 I would suggest arranging the layout with the tables horizontally to make it more streamlined and easier to read.
Response: Thank you so much for your sincere suggestion. We have changed the table into horizontally orientated in the revised version.
-3.4. Add all abbreviations in the table, even if they were already written in the text (e.g., OP, CP, NnP, OnP, NP, RBP, IL, BMI and check the rest).
Response: Thank you so much for your valuable suggestion. We have added all abbreviations in and below the table in this revision (page 11-13).
- Discussion:
-Add to the discussion a comment on the quality of the included studies, most of which were judged to be of low quality.
Response: We appreciate the reviewer for the constructive suggestion. We have added comment on the quality of the included studies in this revision as “However, given by the limited related articles, which were mostly observational/cross-sectional studies, further improvements were indispensable for exploration on the cause-effect relationship. Basically, the overall heterogeneity stated above was somehow unavoidable due to the clinical heterogeneity, including physical conditions of subjects, sample quantity, and clinical methodology (i.e., the aforementioned discrepancy in age, ethnicity, gender, pre-analytical conditions, samples category, sample-collecting and inconsistent measurement methods). All these factors could lead to a conflicting result and reduce the reliability. What’s more, since articles without English publication or incomplete data had been excluded, publication bias was inevitable. If the sample size was sufficient, stratified treatment based on the confounding factors mentioned above would be a better-quality control option. And according to the quality assessment results of two interventional [33, 34] studies based on the ROBINS-I tool, most of domains were considered as low risk and moderate risk when confronting the selection of participants (unmentioned continuous recruitment of patients) or missing data (dropouts of one study [33] were more than 5%). Meanwhile, as for five observational studies [19, 23-26] using the Newcastle-Ottawa Scale tool, the selection and exposure of participants (unmentioned continuous recruitment of patients and discrepancy in BMI threshold for obesity definition due the ethnic groups) were the main concerns that affected the quality. Basically, we only extracted the baseline level of the aimed biomarkers, however, our findings should be interpreted with caution even if so far the overall quality assessment was beyond moderate since the underlying risks and clinical heterogeneity.
All in all, expectance on a deeper and therapeutic investigation of RBP4/visfatin in obese patients affected by periodontitis is meaningful and worthwhile. Given by the currently available evidence from clinical studies, laboratory animal science and in vitro studies, visfatin and RBP4 might exert pro-inflammatory effects, and there is a significant positive correlation between the level of visfatin/RBP4 and the degree of alveolar bone absorption in patients with obesity & periodontitis. Further studies should focus on revealing the role and mechanism of RBP4/visfatin in regulating bone defects in obese periodontitis subjects in vivo. Then, the evaluation of the feasibility of treating alveolar bone absorption in obese patients with periodontitis with RBP4/visfatin monoclonal antibodies or blockers (such as combining novel drug delivery platforms, like injectable gels or microneedles) lays the foundation for long-term clinical transformation and precise treatment.” (page 17-18, line 357-387).
- Conclusion:
-What are the future prospects for clinicians based on these results?
Response: We deeply appreciate your insightful and professional comments. We have modified the manuscript as follows “RBP4 and visfatin might be novel prospective bio-indexes of the comorbidity from diagnosis and prognostic aspect. And the evaluation of the feasibility of treating alveolar bone absorption in obese patients with periodontitis with RBP4/visfatin monoclonal antibodies or blockers lays the foundation for long-term clinical transformation and precise treatment.” (page 19, line 400-404).
- References:
6.1. References in the text are positioned as an index, thus not formatted according to the journal guidelines. Check also the format of the references on line 312.
6.2. References in the bibliography are not formatted according to journal guidelines (e.g., year in bold, volume and journal in italics). Check them all.
Response: Thank you so much for your sincere encouragement and constructive comments, which mean a lot to us. We have changed the format of references according to the journal guidelines.
Round 2
Reviewer 2 Report
Comments and Suggestions for Authors
I congratulate the Authors for improving their manuscript, accepting the reviewers' suggestions and clarifying some doubts.
I add one final comment on this manuscript: the boxes in the PRISMA flow diagram 2020 should not be changed in their order as they reflect the temporal order of the search strategy and study selection. I advise the Authors to re-download the flow diagram and copy only the required fields without inverting any boxes in the image, which must remain unchanged, but obviously filled in. Furthermore, the flow diagram should only present the studies identified by the database search and not also those from the manual search.
Exclusively for explanatory purposes and to help the Authors to fill in the flow diagram correctly, I suggest viewing the flow in the following article:
Di Spirito, F.; Lo Giudice, R.; Amato, M.; Di Palo, M.P.; D’Ambrosio, F.; Amato, A.; Martina, S. Inflammatory, Reactive, and Hypersensitivity Lesions Potentially Due to Metal Nanoparticles from Dental Implants and Supported Restorations: An Umbrella Review. Appl. Sci. 2022, 12, 11208. https://doi.org/10.3390/app122111208
Author Response
Responses to the reviewers’ comments:
We appreciate for your valuable and constructive comments concerning our manuscript. The point-to-point responses are showed below:
Reviewer:
Comments and Suggestions for Authors.
I congratulate the Authors for improving their manuscript, accepting the reviewers' suggestions and clarifying some doubts.
I add one final comment on this manuscript: the boxes in the PRISMA flow diagram 2020 should not be changed in their order as they reflect the temporal order of the search strategy and study selection. I advise the Authors to re-download the flow diagram and copy only the required fields without inverting any boxes in the image, which must remain unchanged, but obviously filled in. Furthermore, the flow diagram should only present the studies identified by the database search and not also those from the manual search.
Exclusively for explanatory purposes and to help the Authors to fill in the flow diagram correctly, I suggest viewing the flow in the following article:
Di Spirito, F.; Lo Giudice, R.; Amato, M.; Di Palo, M.P.; D’Ambrosio, F.; Amato, A.; Martina, S. Inflammatory, Reactive, and Hypersensitivity Lesions Potentially Due to Metal Nanoparticles from Dental Implants and Supported Restorations: An Umbrella Review. Appl. Sci. 2022, 12, 11208. https://doi.org/10.3390/app122111208.
Response: We deeply appreciate your professional suggestions, and apology for misunderstanding your advice at the first place. We have re-download and modified the Figure 1 as the original flow diagram and filled it with our content. Meanwhile, we have deleted the manual search and recorded the processes when we conducted the literature search (page 8).
Thank you very much again for the constructive comments and suggestions. We deeply appreciate the professional comments and hard work you have made to our work.